# Experimental study on the mechanism of surface damage induced by abrasive particles under heavy-load line-contact sliding-rolling conditions

Yang Li[1]*, Wei Wang[2]

**1** Faculty of Engineering, Anhui Sanlian University, Hefei, Anhui, China, **2** Institute of Tribology, Hefei University of Technology, Hefei, Anhui, China

\* 360755791@qq.com

## Abstract

In this study, the JPM-1 contact fatigue testing machine is used to carry out experiments on the mechanism of surface damage induced by abrasive under heavy-load line contact rolling-sliding conditions. In order to be close to the real working conditions, the lubricating oil circuit circulation system is specially designed to ensure that abrasive particles with specific concentration and particle size can continuously circulate in the experiment, so as to simulate the influence of abrasive particles in the lubricating oil on the surface of the friction pair. During the experimental process, with particle size and concentration in the lubricating oil, the rolling-sliding ratio, and the experimental duration as variables, contact fatigue tests are conducted through the coordinated operation of the JPM-1 type testing machine and the lubricating oil circulation system. After the test, KEYENCE VK-X250 series laser microscope is used to observe the surface of the specimen, and combined with 3D topography measurement, the two-dimensional and three-dimensional related parameters are extracted and analyzed. The results show that when the particle size of iron oxide particles increases, the surface roughness and protrusion parameters of the specimen decrease first and then increase, and the volume and number of pits continue to increase. The increase of concentration aggravates the wear, and the roughness parameters, the volume and surface area of the pit convex increase. At the same time, the increase of the slip-roll ratio and the extension of the test time will aggravate the wear, resulting in a significant increase in the coefficient of friction and the amount of wear. This experimental study can provide a reference for the mechanism of surface damage caused by abrasive particles in line contact parts.

## Introduction

The failure of linear contact parts such as gears and bearings caused by abrasive particles in the lubrication system of mechanical equipment is a common

**Data availability statement:** All relevant data are within the paper and its Supporting Information files.

**Funding:** This study received technical support from the Institute of Tribology at Hefei University of Technology and was funded by National Natural Science Foundation of China (51875154), Key Project of Natural Science Research in Colleges and Universities of Anhui Province (2024AH050503) and Anhui Provincial Key Research and Development Program (202004a05020057). The funder provided financial support for the research design, data collection and analysis, and publication.

**Competing interests:** The authors have declared that no competing interests exist.

phenomenon in engineering practice, and has become the main cause of the failure of these parts. In the working process of mechanical equipment, due to the mutual friction between the surfaces of the parts and the invasion of pollutants in the external environment, some wear particles are inevitable in the lubrication system, and with the increase of working time, these wear particles will be more and more, and eventually cause damage and failure of parts. In some special engineering equipment with harsh working environment, the replacement and maintenance cost of bearings, gears and other parts is high, and once these parts are damaged, it will cause large economic losses. Therefore, the study of the situation of abrasive particles entering the linear contact friction surface in the lubrication system, and the in-depth study of the wear mechanism of the linear contact friction interface by using the testing machine and combining a variety of observation and analysis methods can effectively avoid the failure of the linear contact parts.

At present, the research on the failure analysis of abrasive-induced heavy-duty linear contact parts presents diversified characteristics in technical methods, mainly covering multiple dimensions such as microstructure analysis, chemical composition analysis and mechanical property evaluation. In terms of microstructure analysis, Mohammad A [1] used a ZEISS AX10 optical microscope and the Dutch Phenom Pharos G2 desktop field emission scanning electron microscope (FE-SEM) to observe the microscopic morphological changes of deterioration phenomena such as abrasive wear, revealing micro-damage characteristics such as grooves and micro-cracks formed during the surface scratching process of wear debris. Panin Sergey V et al. [2] used focused ion beam (FIB) technology to precisely cut and prepare the wear region of the friction pair of the Mo-W-S-Se composite film, and then used transmission electron microscopy (TEM) to analyze the microscopic interface of the interaction, which provided a reference for the microscopic experimental method for analyzing the damage mechanism of abrasive particles to the surface of the friction pair. In addition to the analysis of microstructure, chemical composition analysis is also one of the important means to study the damage mechanism of abrasive particles to linear contact surfaces. Laser-Induced Breakdown Spectroscopy (LIBS) was used to perform rapid elemental analysis of abrasive particles in the lubricating oil of the friction pair, and the wear type and damage degree of the friction pair surface under different working conditions were inferred by detecting the changes in the content of different elements in the abrasive particles [3–5]. Several scholars have demonstrated that friction and abrasion occur on a solid surface when it comes into contact with another solid surface under pressure and slides, and in some cases, a thin layer of powdery wear debris adheres to the worn surface, X-ray photoelectron spectroscopy (XPS) is used to analyze the chemical composition of the abrasive contact surface, which can reveal the composition of wear debris and provide chemical information for in-depth understanding of the damage mechanism of abrasive particles to the surface [6–8]. The evaluation of mechanical properties focuses on the influence of abrasive particles on the mechanical properties of linear contact surfaces. Su Y et al. [9] carried out friction and wear experiments with relative

speed, load, abrasive particle size and abrasive mass fraction as the process parameters through a self-developed free abrasive grinding experimental device under line contact conditions, and studied the evolution law of friction coefficient and wear amount of cast iron materials under different process parameters, providing a basis for analyzing the changes in mechanical property parameters such as hardness and elastic modulus before and after abrasive particles act on line-contact surfaces. Samadani A et al. [10] developed a model to predict the coefficient of friction in the mixed lubrication state. The model can effectively predict the friction coefficient in the presence of lubricant in the line contact problem, so as to indirectly reveal the mechanical properties of the friction pair interface. Zhaoming Yin et al. [11] investigated the morphological characteristics of pitting-induced wear particles through numerical simulation of gear contact fatigue pitting, providing theoretical insights for establishing a mechanical model that correlates line-contact wear particle features with wear state characterization. These research methods of different dimensions complement each other and jointly promote the in-depth development of the research on the mechanism of abrasive-induced surface damage under line contact conditions.

In this paper, the simulation test of wear particles entering the linear contact friction surface in the lubrication system is systematically designed, and a series of linear contact friction surface wear tests are carried out, and the influence of different test conditions on the wear of the linear contact friction surface is analyzed by changing the particle size, concentration of iron oxide particles, slip-roll ratio and time of the test. The causes of failure of heavy-duty line contact parts induced by abrasive particles in the lubrication system are analyzed and detected, and the influence of wear particles on the surface wear of the specimens is quantitatively analyzed, so as to put forward suggestions for improvement. In this paper, the influence of different experimental conditions on the wear of linear contact friction surface is analyzed, the JPM-1 contact fatigue wear testing machine is used to simulate the heavy-duty linear contact slip working condition, and the optical observation, 3D topography measurement, two-dimensional and three-dimensional related parameters extraction of the surface of the specimen are carried out by KEYENCE's VK-X250 series shape measurement laser microscope and its supporting analysis software. The wear phenomenon of linear contact friction surface affected by abrasive particles is studied. This study provides a theoretical basis for the mechanism of abrasive-induced surface damage in the slip-roll state of heavy-duty line contact.

## Experiments and methods

### Preparation of lubricating oil samples for experiments

The lubricating oil is Great Wall 32# basic machinery lubricating oil (GB 443–1989). The content of abrasive particles in the lubricating oil in the test is selected according to the classification standard of oil pollution degree and the grade of oil pollution degree in the actual lubrication system. Since the average contamination degree of the transmission oil is 23/20/15, the concentration of particles in the test oil is 0.4 mg/L and 4 mg/L for the control test. Most of the abrasive particles in normal transmission fluid were between 1 and 50 microns, and the distribution was most concentrated around 7 microns [12,13], so five iron oxide particles of 1 micron, 5 microns, 10 microns, 18 microns and 28 microns are used for control experiments. The microscopic shape and particle size distribution of the iron oxide particles used in the test lubricating oil have a great influence on the test results, so it is necessary to observe them and analyze the particle size. The scanning electron microscope is used to observe the iron oxide particles with a particle size of 10 microns, and the specific observation images are shown in Fig 1, through which it can be seen that the initial shape of the iron oxide particles with a particle size of 10 microns is almost spherical, and the spherical particles are extruded and deformed during the test process, which makes it easier to form grooves on the surface of the specimen. At the same time, the particle size analyzer is used to detect the particle size distribution of iron oxide particles, and the specific detection results are shown in Fig 2. Table 1 is the volume distribution of particle size in each range, in which the particle size varies from 0.1 microns to 40 microns, which is more consistent with the particle size distribution in SEM observation.

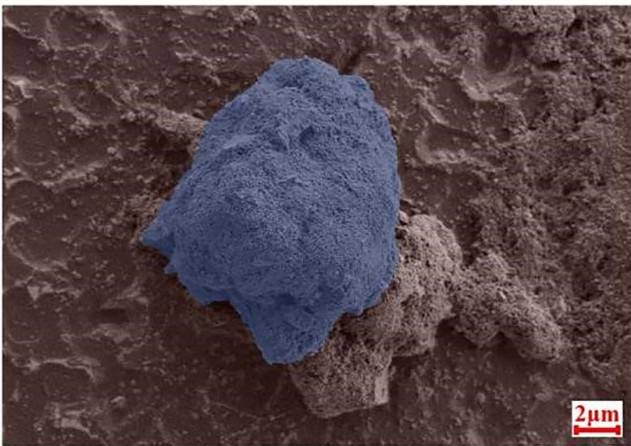

**Fig 1. SEM image of 10-micron iron oxide particles.**

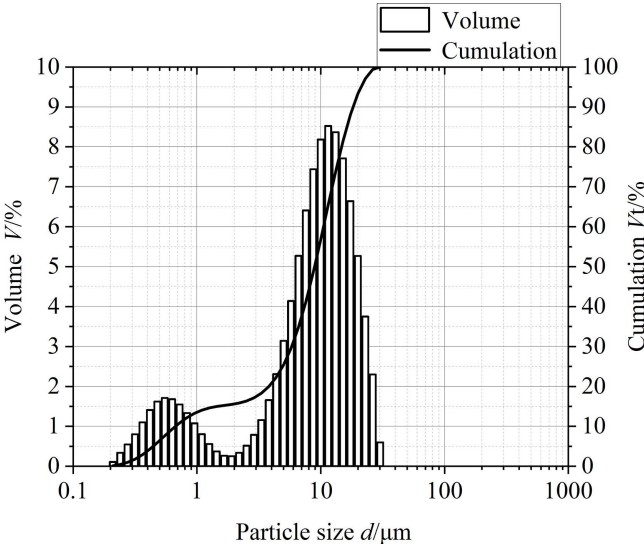

**Fig 2. Particle size distribution chart of 10-micron iron oxide particles.**

**Table 1. Particle size distribution of iron oxide.**

| Particle size/μm | Volume in range/% | Particle size/μm | Volume in range/% |
|---|---|---|---|
| 0.01~0.10 | 0 | 5.00~10.00 | 34.59 |
| 0.10~1.00 | 13.26 | 10.00~20.00 | 36.51 |
| 1.00~5.00 | 9.00 | 20.00-40.00 | 6.64 |

## Experimental methods

The experiment on the surface damage mechanism induced by abrasive particles under the condition of heavy load line contact slip is carried out on the JPM-1 contact fatigue wear testing machine. The specimen material is made of 40Cr, which is commonly used in the manufacture of gears, bearings and other parts, and the hardness of the 40Cr specimen

reaches 50~60HRC through heat treatment, which is in line with the surface contact strength of most key components in mechanical equipment. The specimen is installed as shown in Fig 3, the lower specimen is an original specimen with an inner circumference diameter of 30 mm, an outer circumference diameter of 60 mm, and a width of 20 mm, in order to increase the Hertzian contact pressure between the specimens in the experiment to make it more in line with the actual working conditions, the upper specimen is designed into a stepped shape to reduce the contact length between the two specimens and thus increase the Hertz contact pressure, so the upper specimen is obtained by turning the original specimen with a length of 7.5 mm and a depth of 3 mm along the axial direction from the end to the middle. The contact length of the two specimens is 5 mm. Before the test, all the specimens are finely ground to obtain a more uniform initial surface morphology. The surface optics and three-dimensional topography of the initial specimen are shown in Fig 4, and the specific relevant dimensional performance parameters are shown in Table 2. In order to make the maximum Hertz contact pressure of the two specimens reach more than 2GPa under the normal operation of the machine, the test load is selected to be 10kN after calculating according to the Hertz contact pressure calculation formula [14].

## Experimental lubricating oil circuit transformation

In order to simulate the lubrication of mechanical parts under real working conditions, a lubricating oil circuit circulation system as shown in Fig 5 was designed on the experimental machine. This oil circuit circulation system consists of a peristaltic pump, a stirrer and a funnel connected through the pipeline, the agitator mixes the experimental lubricating oil and particles and stir, the oil wraps the particles with the rotation of the upper specimen and enters the interface of the two

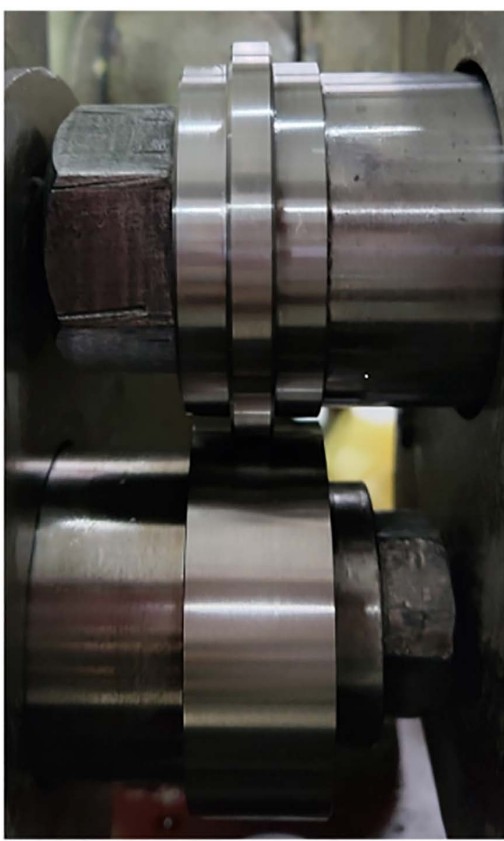

**Fig 3. Installation drawing of experimental test specimen.**

a

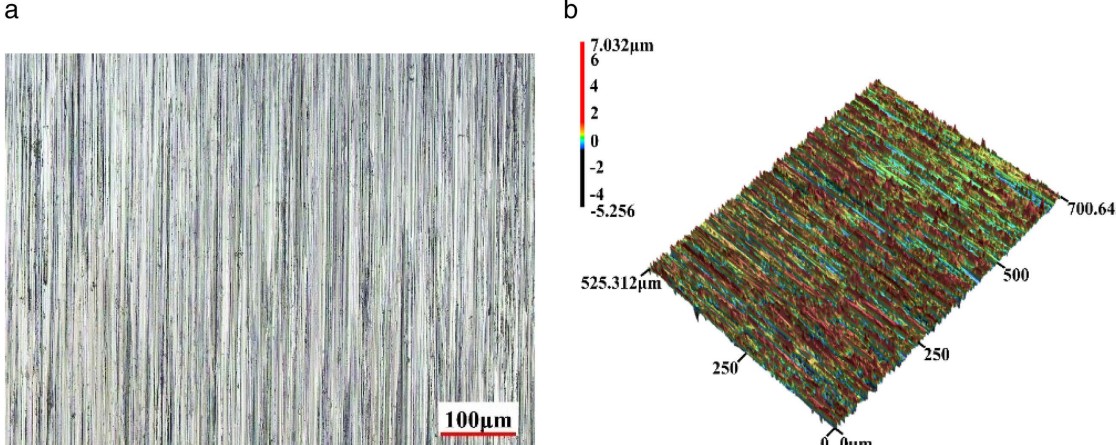

b

**Fig 4. (a). Initial microscopic morphology map of test specimen surface. (b). Initial three-dimensional morphology map of test specimen surface.**

**Table 2. Relevant parameters of test piece.**

| parameter | numeric value | parameter | Value/μm |
|---|---|---|---|
| Material type | 40Cr | Size/mm | F60×30×20 |
| Coaxiality/mm | 0.05 | Contact length/mm | 5 |
| Hardness/HRC | 50-60 | Surface roughness/Ra | 0.612 |
| Modulus of elasticity/GPa | 206 | Root mean square/Rq | 0.756 |
| Poisson's ratio | 0.28 | Maximum height/Rz | 4.948 |

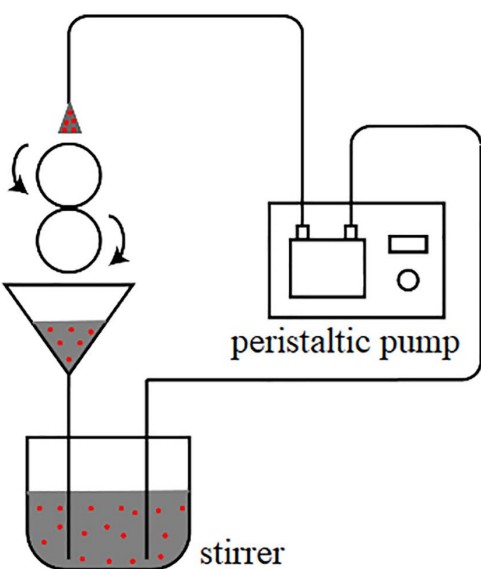

**Fig 5. Test oil circuit circulation diagram.**

specimens to squeeze and rub against each other for three-body friction, and then the oil drops into the funnel with the rotation of the lower specimen, and finally flows into the agitator through gravity, so as to realize a complete set of circulation of oil in the test.

## Experimental plan and experimental process

This study examines the wear characteristics of specimen surfaces by adjusting iron oxide particle size, concentration, slide-to-roll ratio, test duration, and the influence of different conditions on the surface wear of the specimen is explored by means of optical observation and three-dimensional morphology analysis and measurement of the worn surface. The specific pilot plan is shown in Table 3.

## Introduction to experimental instruments

In this paper, the JPM-1 contact fatigue testing machine is selected to simulate the rolling contact fatigue mechanism induced by abrasive particles under heavy-duty linear contact rolling conditions, this testing machine adopts a friction pair configuration in which two cylindrical specimens (upper and lower) rotate against each other in either pure rolling or rolling-sliding contact. and simulates the contact fatigue and wear of mechanical parts (such as gears, bearings, rolls, rails, etc.) in linear contact during operation.

In order to explore the influence of abrasive particles on the surface damage mechanism of the friction pair under the lubrication condition of the real linear contact parts, a modified lubricating oil circulation system is used to simulate the influence of the lubricating oil containing abrasive particles on the working conditions of the linear contact experimental surface.

In addition, KEYENCE's VK-X250 series shape measurement laser microscope and its supporting analysis software are used to perform optical observation, 3D topography measurement, and 2D, 3D correlation parameter extraction on the surface of the specimen. This is a set of optical microscope, SEM, roughness analyzer and other equipment functions in one of the multi-functional microscopic analysis instrument, the instrument has two light sources of white and laser, when observing the test piece, it can be generally optical observation, can also use the laser in X, Y, Z three directions to scan the surface of the specimen for measurement, so as to obtain the three-dimensional morphology of the surface to be measured. The observation images obtained by this instrument have high resolution, excellent overall image quality, simple

**Table 3. Experiment Schedule.**

| The name of the experiment | Experimental variables | | Experimental conditions |
|---|---|---|---|
| Effect of iron oxide particle size in lubricating oil on specimen surface | Particle size (µm) | 0 | Room temperature, load is 10kN, speed is 350 rpm, slip-roll ratio is 15%, test time is10h, iron oxide concentration in oil solution is 0 mg/mL, 0.4 mg/mL and 4 mg/mL. |
| | | 1 | |
| | | 5 | |
| | | 10 | |
| | | 18 | |
| | | 28 | |
| Effect of iron oxide concentration in lubricating oil on the surface of the specimen | Concentration (mg/mL) | 0 | Room temperature, load is 10kN, the speed is 350 rpm, the slip-roll ratio is 15%, test time is 10h, and particle size of iron oxide in the oil solution is 1µm, 5µm, 10µm, 18µm and 28µm. |
| | | 0.4 | |
| | | 4 | |
| Effect of slip-roll ratio on the surface of the specimen | Slip-roll ratio | 5% | Room temperature, load is 10kN, t speed is 350 rpm, test time is 10h, particle size of iron oxide in the oil solution is 0, and concentration is 0. |
| | | 15% | |
| | | 30% | |
| The effect of test time on the surface of the specimen | Time (h) | 10 | Room temperature, load is10kN, speed is 350 rpm, slip-roll ratio is 5%, iron oxide particle size in oil solution is 0, concentration is 0. |
| | | 30 | |
| | | 60 | |
| | | 100 | |

operation, fast measurement, and can capture a variety of complex surface shapes, and quickly obtain accurate and reliable data. The measurement results can be measured and quantitatively analyzed by using the supporting VK-X Series multi-file analysis software to measure the surface profile and shape, and the measurement data of the two-dimensional and three-dimensional surface topography parameters related to single lines, multiple lines, surface roughness, surface contours, pits and protrusions can be obtained. The instrument can be used to observe the microscopic morphology of the specimen surface and measure the surface parameters of the specimen before and after the test. In this way, the surface damage mechanism of abrasive particles-induced heavy load line contact parts can be more accurately analyzed.

## Experimental results and analysis

### Analysis of the influence of particle size in lubricating oil on the surface of the specimen

Fig 6 and Fig 7 are numerical diagrams of the main surface roughness parameters of the specimens after the test with different particle sizes at concentrations of 0.4 mg/mL and 4 mg/mL, respectively. It can be seen from Fig 6 and Fig 7

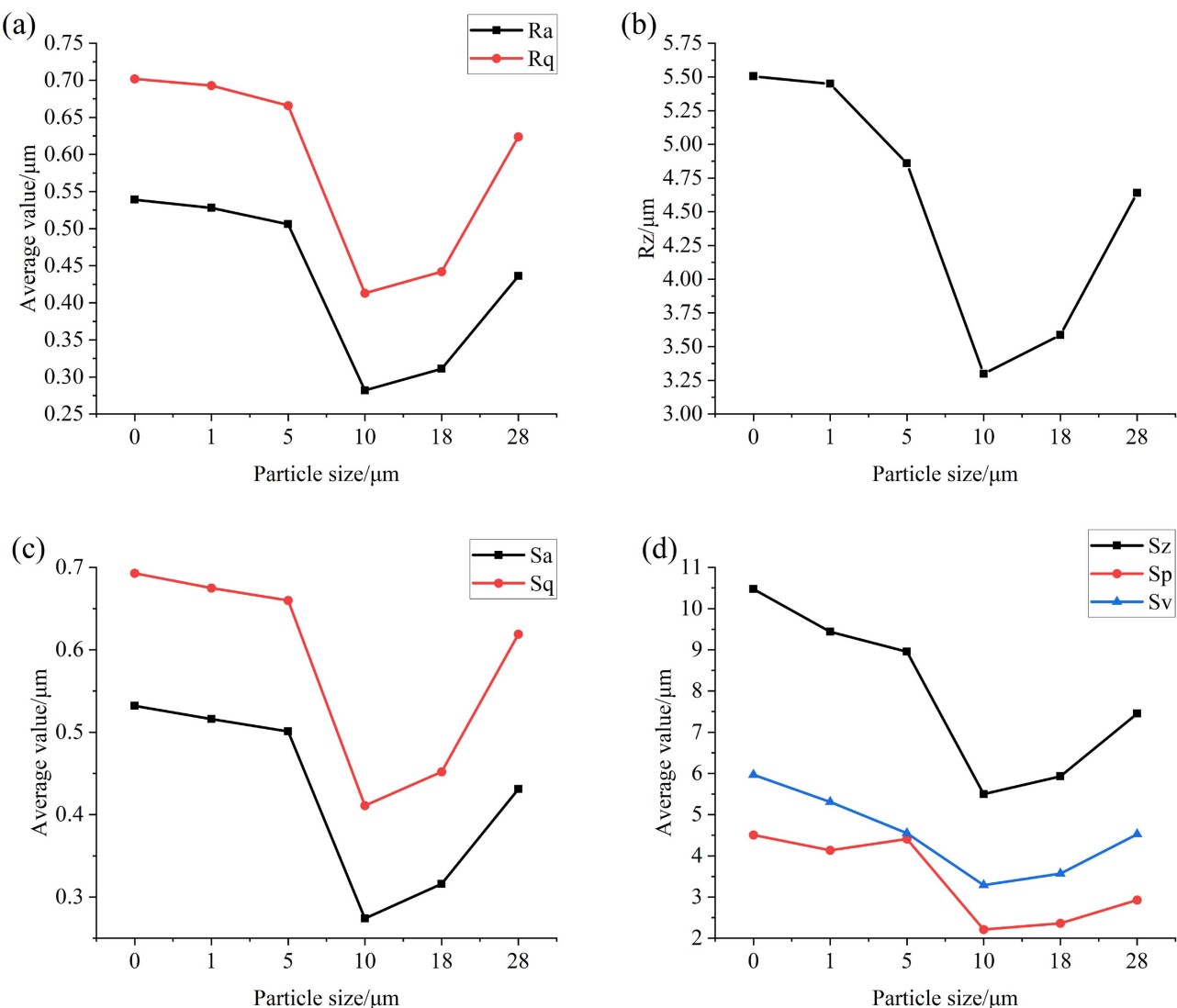

**Fig 6. Graph of surface roughness values of specimens after testing with different particle sizes at a concentration of 0.4 mg/mL (a) Ra, Rq (b) Rz (c) Sa, Sq (d) Sz, Sp, Sv.**

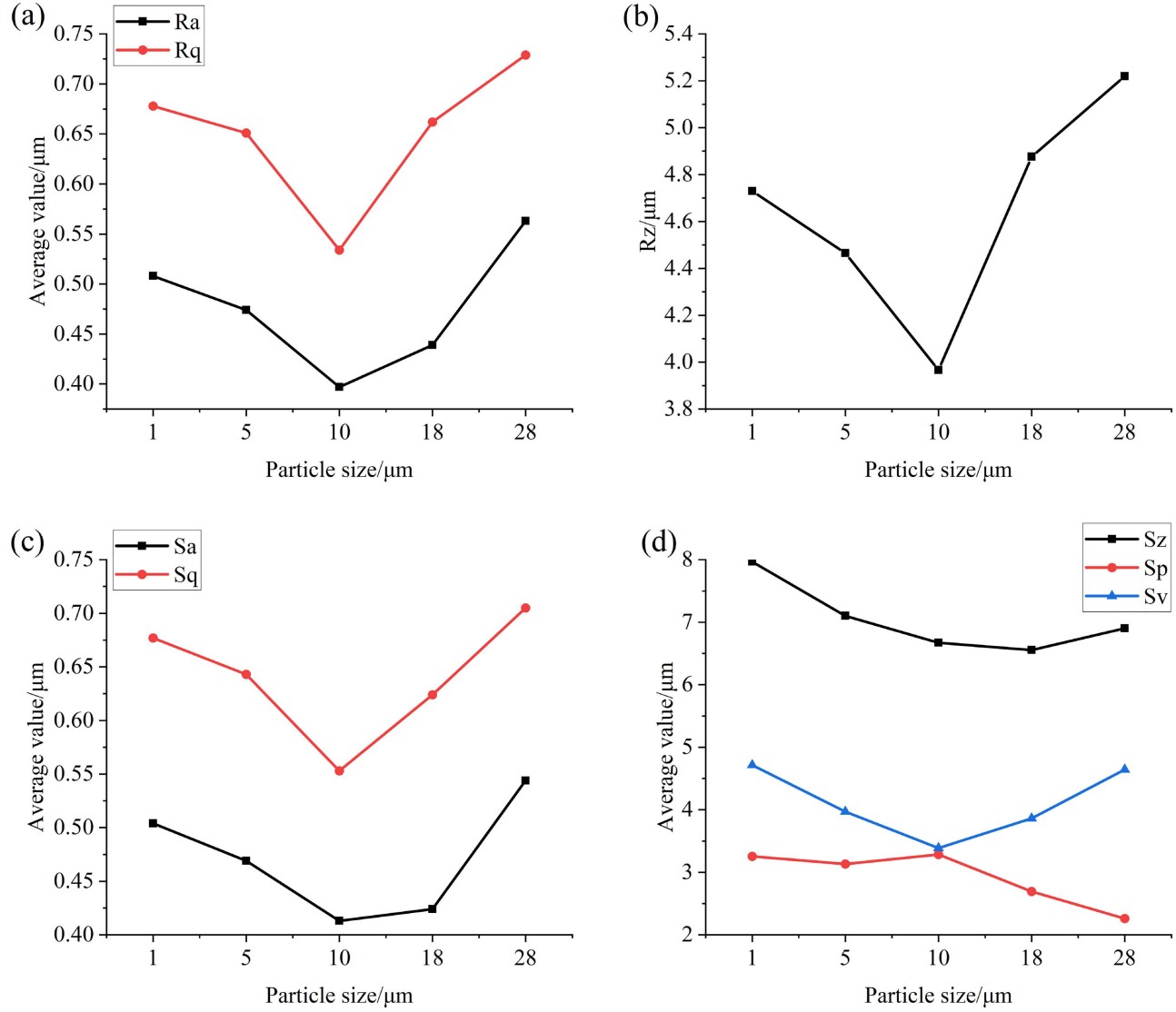

**Fig 7. Graph of surface roughness values of specimens after testing with different particle sizes at a concentration of 4 mg/mL (a) Ra, Rq (b) Rz (c) Sa, Sq (d) Sz, Sp, Sv.**

that when low concentration and small particle size iron oxide particles are added, the surface roughness values of the specimen Ra, Rq, Rz, Sa, Sq, Sp, Sv and Sz all decrease after the test, because when the oil contains particles with low concentration and small particle size, the particles enter the friction surface with the oil, and the particle size is greater than the thickness of the oil film, the average height and distribution inhomogeneity of asperities on the specimen surface are reduced under sliding-rolling motion, and the relative pit depth also decreases, so that the surface tends to be flatter, so that the values of Ra, Rq, Rz and Sa, Sq, Sp, Sv and Sz gradually decrease. When the particle size of the particles in the oil is greater than 10 μm, the effect of the particles on the friction surface wear is intensified, so that more micro-convex bodies and pits continue to appear on the relatively flat surface in the test, and the height and pit depth of these micro-convex bodies gradually increase with the increase of particle size, and the unevenness of distribution is gradually strengthened, so that the values of Ra, Rq, Rz, Sa, Sq, Sp, Sv and Sz gradually increase.

After the experiment, the volume and area measurement function of the surface topography analysis module in the VK analysis software is also utilized to extract and analyze the characteristic parameters of the convex and concave regions on the specimen surface. In order to avoid the influence of the curvature of the specimen itself, the surface shape of the quadric surface is corrected, and the part 0.3 μm higher than the datum is selected as the convex area, and the part 0.3 μm lower than the datum is selected as the concave area. The relevant parameters of the convex and depressed areas on the surface of the specimen after the test of particles of different particle sizes at a concentration of 0.4 mg/mL are shown in Fig 8 and Fig 9, in which the main measured parameters are the average volume, total volume, average surface area, total surface area, average height, maximum height, average cross-sectional area of the convex area in the convex area, and the average volume, total volume, average surface area, total surface area, average depth, maximum depth and average cross-sectional area of the pit in the pit area. It can be seen from the figure that when the particle size is small, the maximum height of the convex part on the surface of the specimen is basically unchanged after the test, the average height fluctuates, but the overall change is not large, the average volume, the total volume, the average

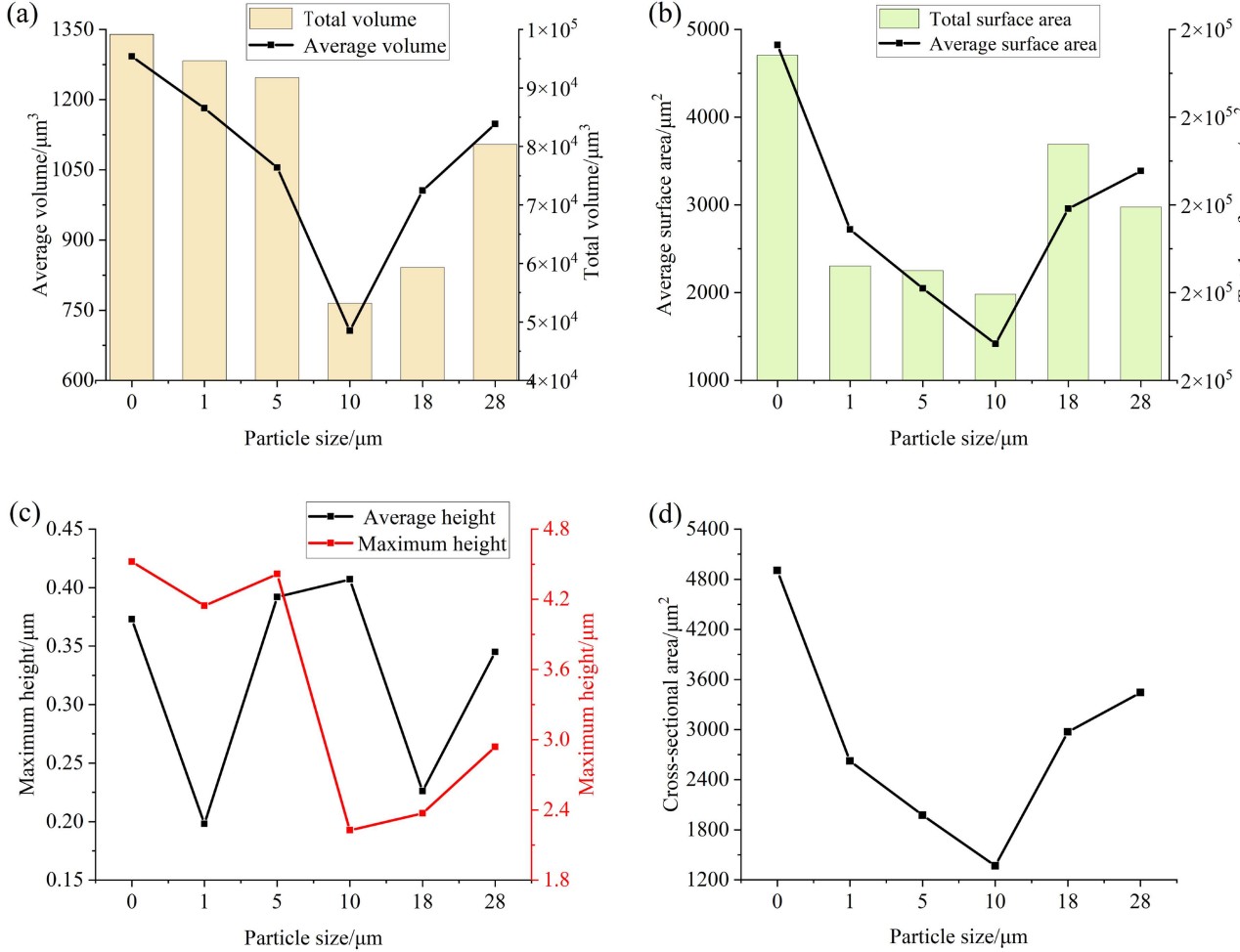

**Fig 8. Parameter diagram for the raised parts on the surface of the test specimen (a) Volume (b) Surface area (c) Height (d) Cross-sectional area.**

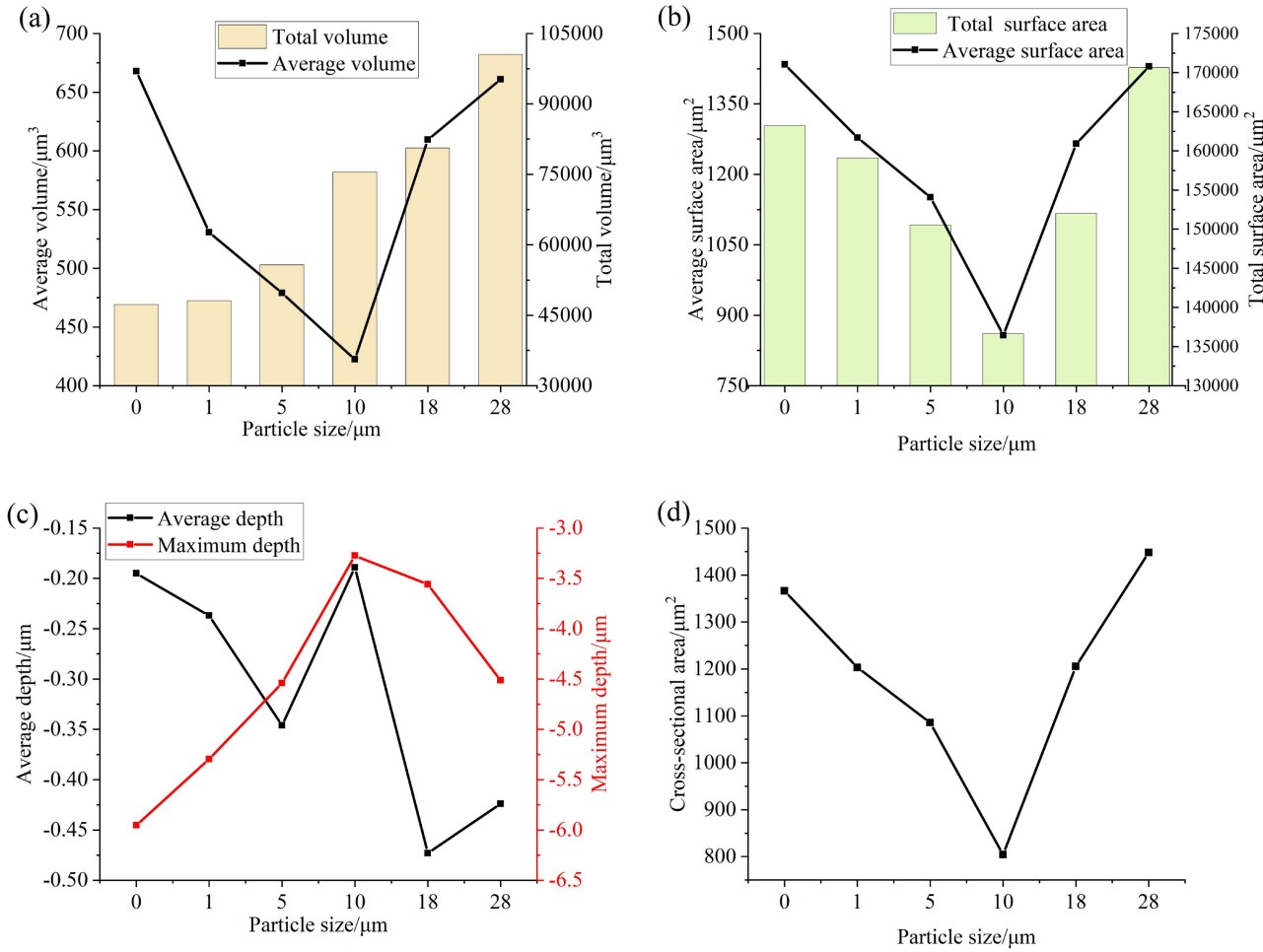

**Fig 9. Parameter diagram for the pit parts on the surface of the test specimen (a) Volume (b) Surface area (c) Depth (d) Cross-sectional area.**

surface area, the total surface area and the average cross-sectional area all show a downward trend, and the maximum height of the protrusion and the total volume decrease greatly at 10 μm, and the maximum height, average volume, average surface area, total surface area and cross-sectional area of the pit part also conform to this trend. However, the average depth and total volume of the pits are gradually increasing, because when the particle size is small, the particles between the friction surfaces are squeezed and sliding, which accelerates the wear of the convex part on the surface of the specimen, thereby reducing the parameter values of the convex part, but due to the plastic deformation of the particles on the friction surface, the abrasive wear occurs on the surface of the specimen, so that the depth and total volume of the pits increase, and the decrease of the average volume of the pits indicates that the overall number of pits is increasing. When the particle size is greater than 10μm, the parameters of the convex and pit on the surface of the specimen increase, indicating that the surface wear of the specimen is aggravated, which is due to the fact that the particles with larger particle size will break and produce more small particles in the process of being squeezed by the specimen, which will produce greater stress in this process, resulting in the loss of the surface material of the specimen and the increase of the wear area.

**Analysis of the influence of particle concentration in lubricating oil on the surface of the specimen**

Fig 10 is a numerical comparison of the main surface roughness parameters of the specimen after the test of different particle concentrations in each particle size group, and it can be seen from the figure that the change trend of the main surface roughness parameters of the specimen after the test with a concentration of 4 mg/mL is basically consistent with the change trend of the main surface roughness parameters of the specimen after the test with a concentration of 0.4 mg/mL, and both show a trend of first decreasing and then increasing. Moreover, the surface roughness parameter value of the specimen containing high concentration particle test is smaller than that of the specimen surface roughness parameter value after the low concentration test when the particle size is small, but when the particle size gradually increases, the surface roughness parameter value of the specimen after the high concentration test is

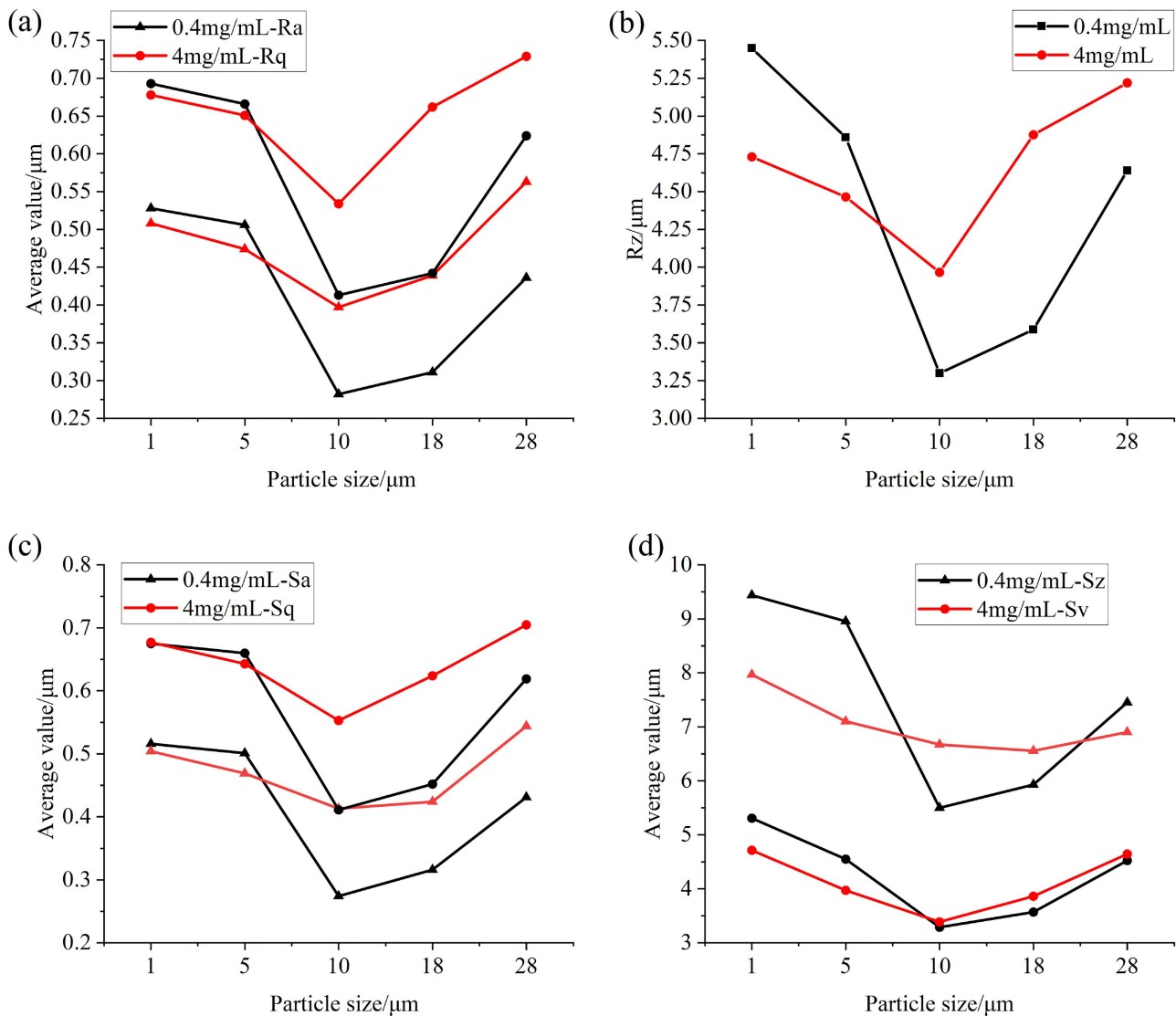

**Fig 10. Graphs of surface roughness values of test specimens after testing with varying concentrations across different particle size groups (a) Ra, Rq (b) Rz (c) Sa, Sq (d) Sz, Sv.**

larger than that of the low concentration test. This is due to the increase of particle concentration in the oil, resulting in the increase of the number of particles entering the friction interface, when the particle size is small, the particles act on the friction surface more to accelerate the wear of the micro-convex body, so that the surface becomes more flat, and also accompanied by the occurrence of abrasive wear, when the particle size increases, the surface wear is aggravated, resulting in more micro-convex bodies and pits, resulting in the increase of surface roughness parameter value.

Fig 11 and Fig 12 are the comparison of the total volume, total surface area, average height, average depth and average cross-sectional area of the convex and pit areas on the surface of the specimen after each particle size test at different concentrations. It can be seen from the figure that the parameter change trend of the convex and pit parts

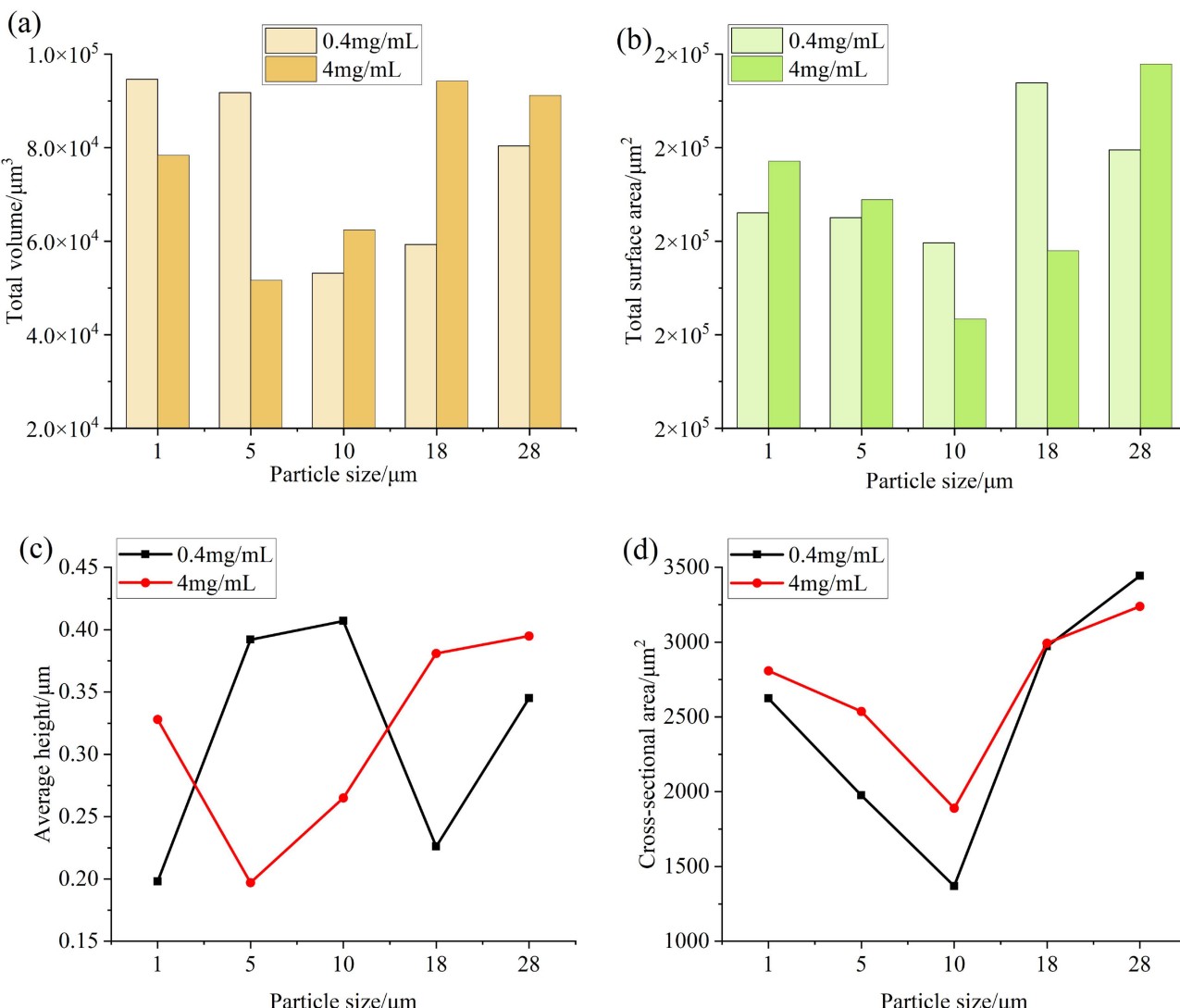

Fig 11. Parameter diagram for the raised portions on the surface of the test specimen (a) Total volume (b) Total surface area (c) Average height (d) Cross-sectional area.

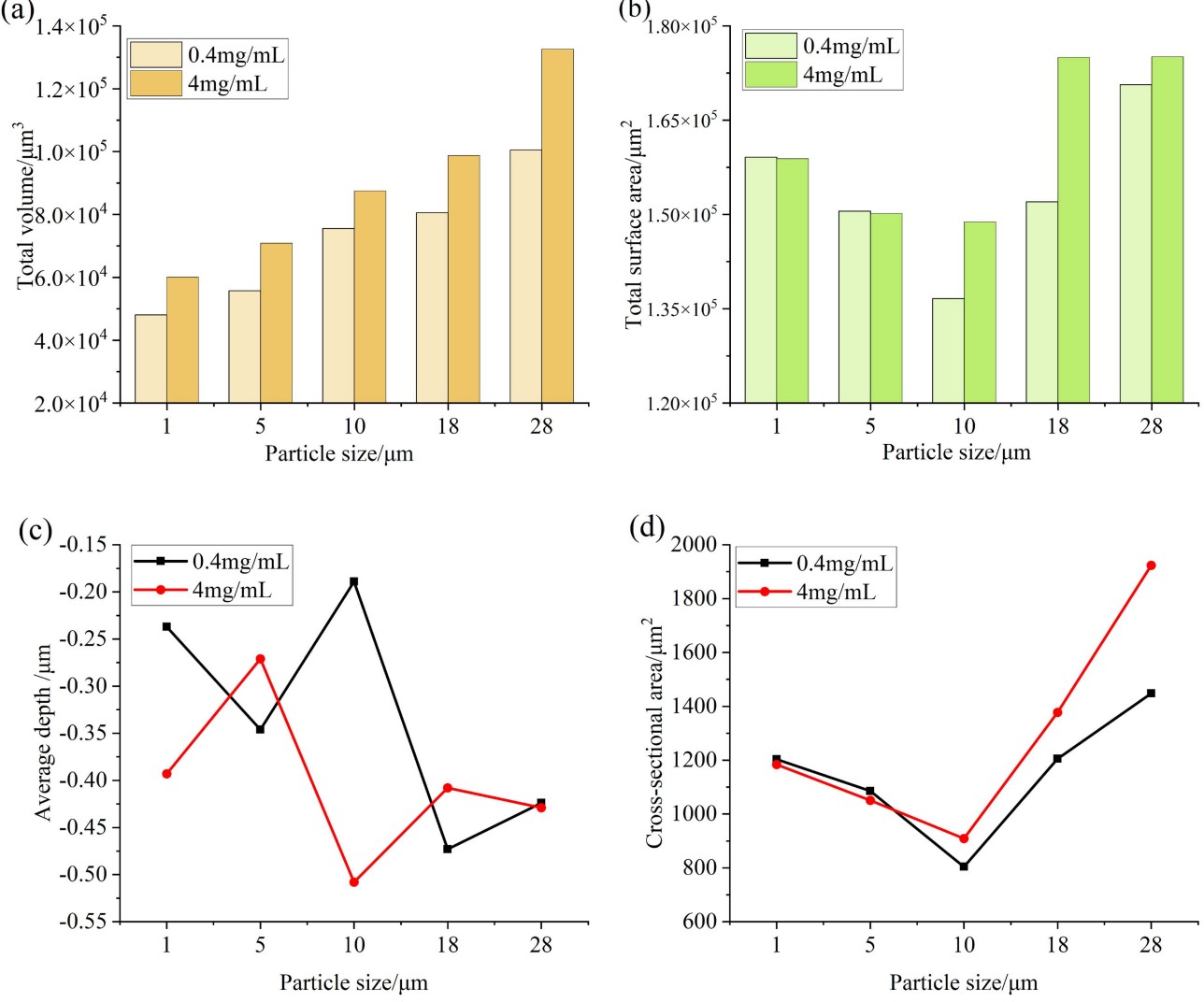

**Fig 12. Parameter diagram for the pit portions on the surface of the test specimen (a) Total volume (b) Total surface area (c) Average depth (d) Cross-sectional area.**

on the surface of the specimen at high concentration is basically consistent with the change trend at low concentration, and the total volume and total surface area of the protruding part decrease first and then increase with the increase of particle size, and the decrease and increase range under high concentration conditions are higher than those at low concentration, while the average height and cross-sectional area do not change much. The total volume of the pit increases with the increase of particle size, while the total surface area and cross-sectional area of the pit decrease first and then increase, and the total volume of the pit is larger than that at low concentration under the condition of high concentration, and the increase is also increasing with the increase of particle size, and the total surface area and cross-sectional area of the pit also increase with the increase of particle size, and the total surface area and cross-sectional area of the pit also increase compared with the total surface area and cross-sectional area of the pit at low concentration.

## Analysis of the influence of the slip-roll ratio on the surface of the specimen

As shown in Fig 13, it can be seen that when the slide-to-roll ratio increases, the multi-line roughness value and surface roughness value and root mean square value of the specimen surface increase, which indicates that when the slide-to-roll ratio increases, the sliding distance increases, and the number of rough stress cycles increases, resulting in an increase in the height and distribution of the micro-convex body on the surface of the specimen.

Fig 14 and Fig 15are the relevant parameters of the convex part and the pit part on the surface of the specimen after the test under different slip-roll ratios, and the relevant treatment steps are consistent with the particle size-concentration test. It can be seen from the figure that with the increase of the slide-to-roll ratio, the total volume and average height of the convex part on the surface of the specimen are increasing, while the average volume, total

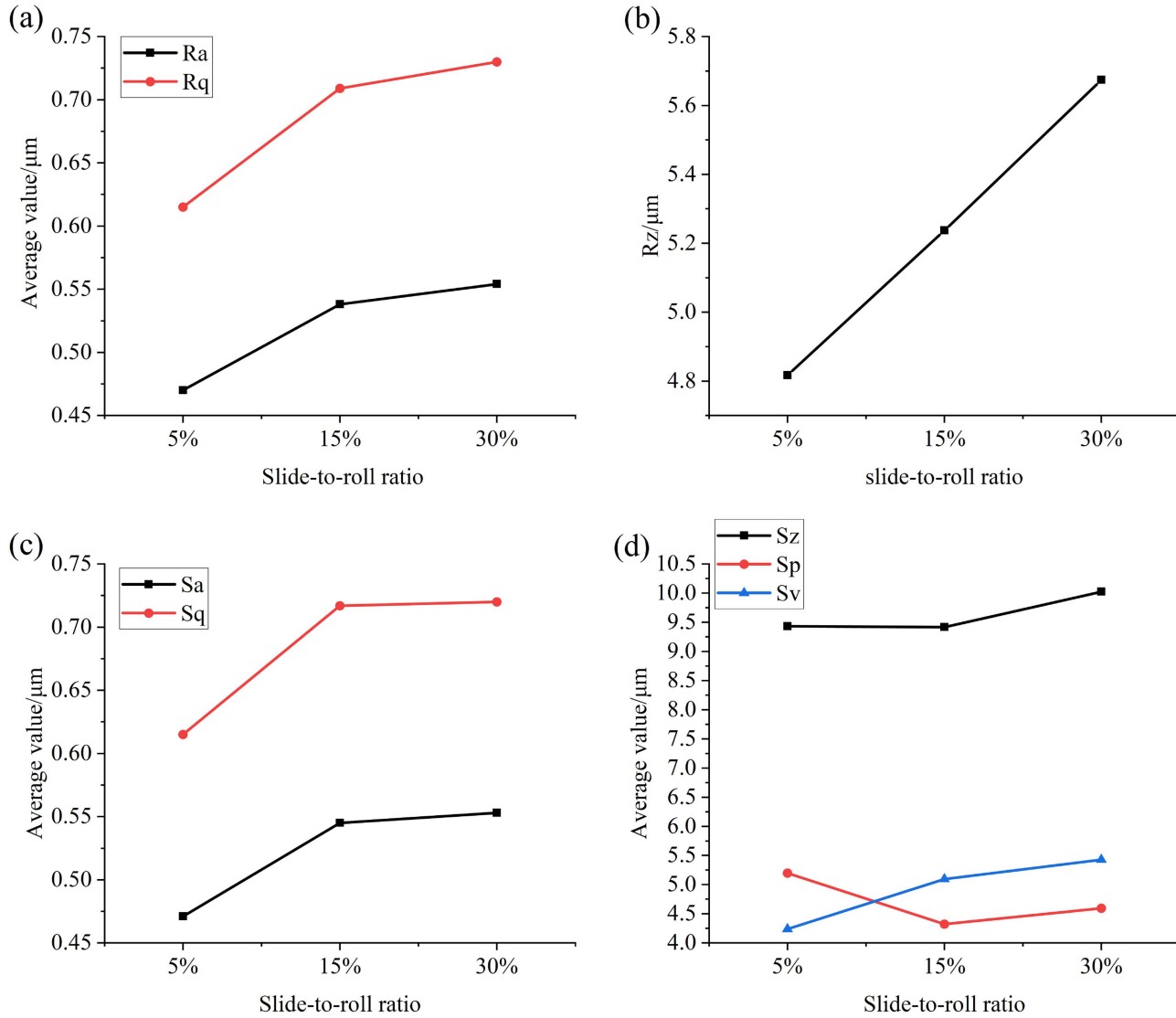

**Fig 13. A graph showing the surface roughness values of the test specimen after tests with different slide-to-roll ratios (a) Ra, Rq (b) Rz (c) Sa, Sq (d) Sz, Sp, Sv.**

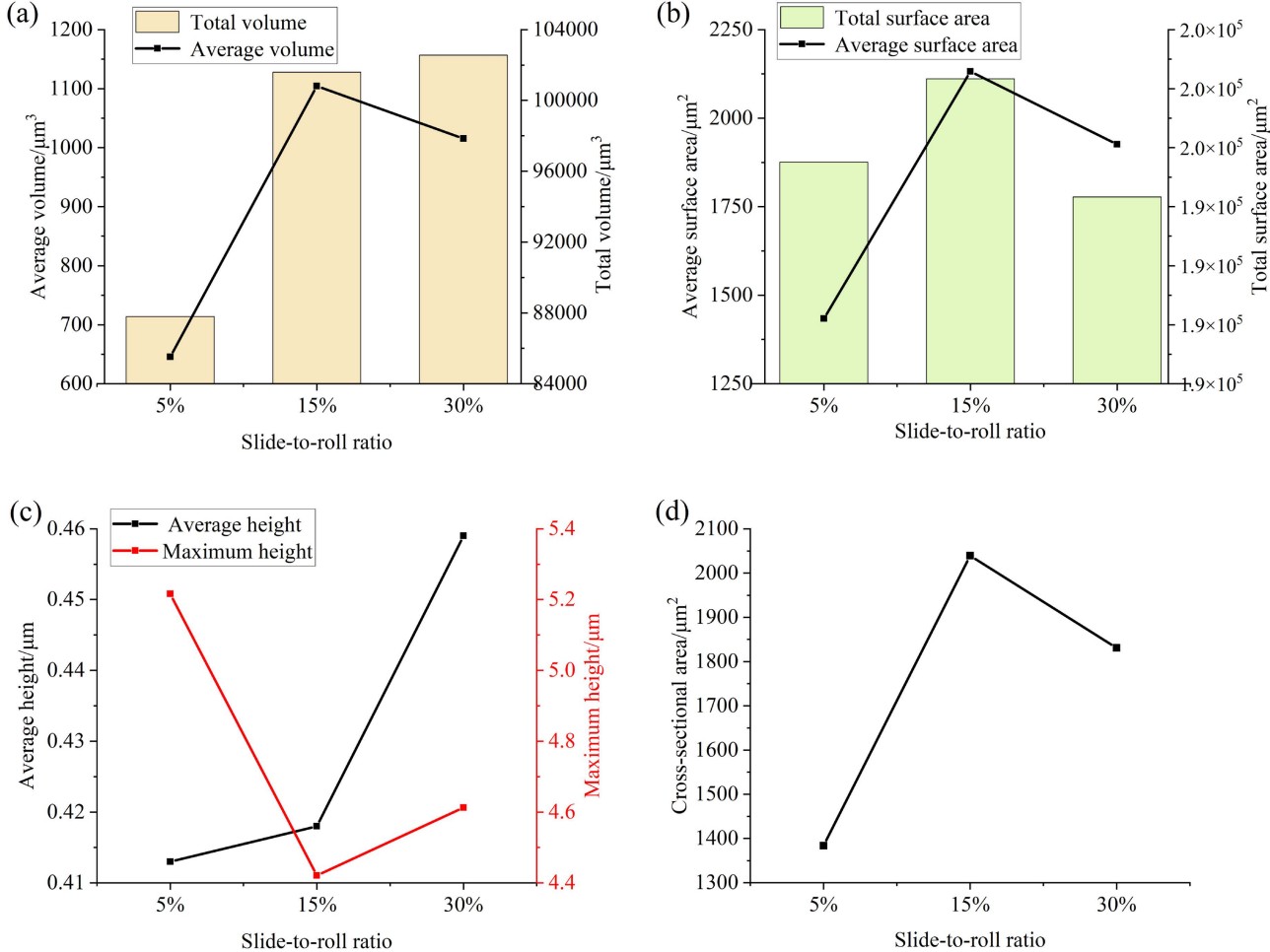

**Fig 14. Parameter diagram for the raised portions on the specimen surface (a) Volume (b) Surface Area (c) Height (d) Cross-sectional area.**

surface area, average surface area and cross-sectional area increase first and then decrease, but the decrease is small. At the same time, the total volume, average volume, total surface area and average depth of the pit part all increase with the increase of the slide-to-roll ratio, which indicates that the overall wear of the surface of the specimen is aggravating, the number of protrusions and pits is increasing, and the surface damage is becoming more and more serious, because the higher slip would increase the number of stress cycles during each contact, thereby promoting the occurrence of surface damage.

## Analysis of the influence of experimental time on the surface of the specimen

Fig 16 is the numerical diagram of the two-dimensional and three-dimensional roughness of the surface of the specimen under different test times, as shown in the figure, when the test time increases, the values of multi-line roughness Ra, Rq, Rz and the values of surface roughness Sa, Sq are gradually increasing, and the more the test time, the greater the increment, which shows that when the test time is small, the specimen under the condition of net oil lubrication is in good working condition, and the wear on the surface of the specimen is less, and with the increase of the test time, the surface wear is aggravated. As a result, the clean lubricating oil is mixed with the particles worn down on the surface of the specimen,

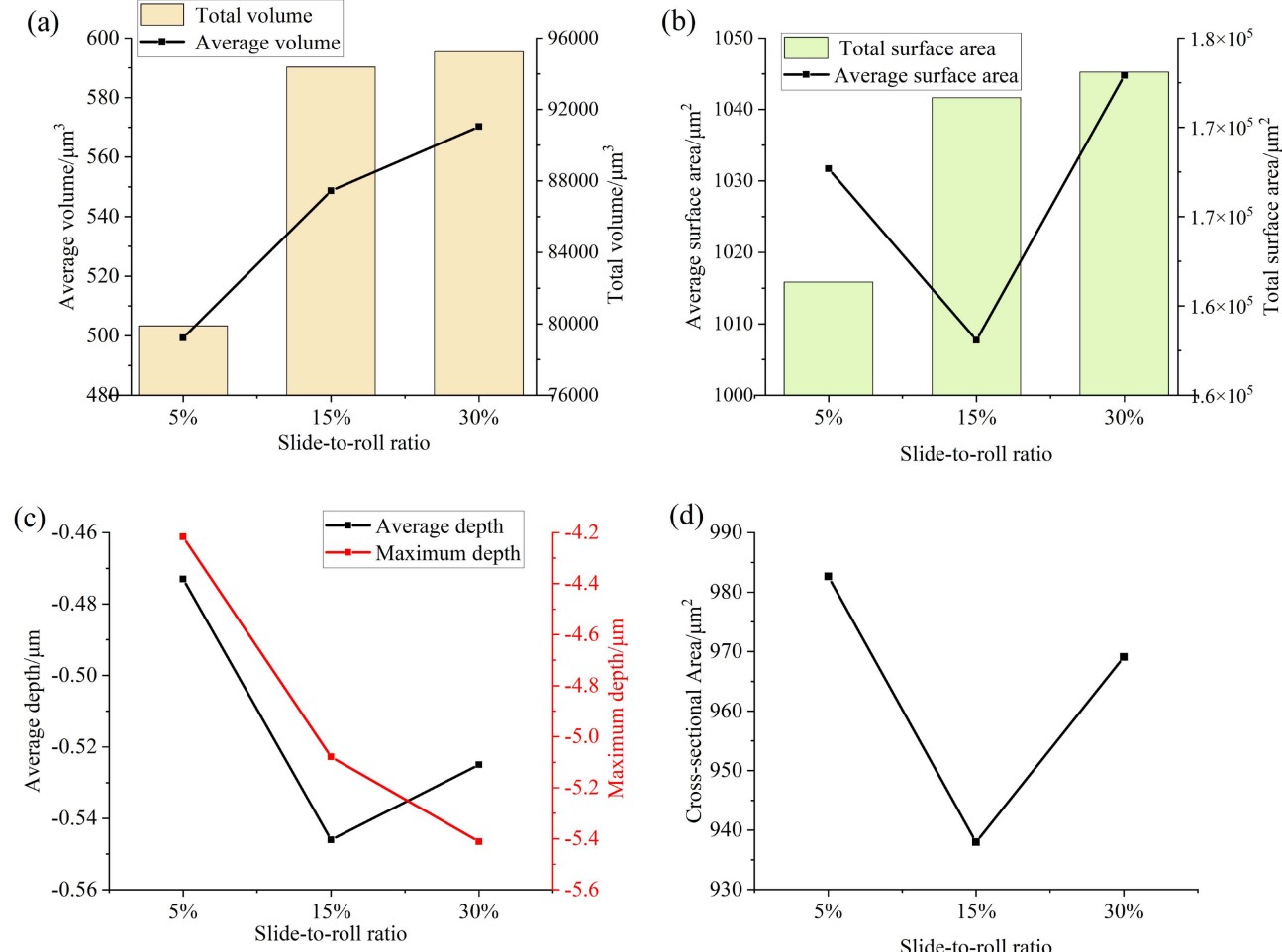

**Fig 15. Parameter diagram for the pit portions on the specimen surface (a) Volume (b) Surface Area (c) Depth (d) Cross-sectional area.**

and these particles further accelerate the wear as the oil enters the friction surface, resulting in a further increase in the roughness value of the specimen surface.

Consistent with the analysis steps obtained in the previous test, Fig 17 and Fig 18 are the relevant parameters of the convex and pit parts on the surface of the specimen after different test times. It can be seen from the figure that when the test time is small, the total volume and total surface area of the convex and pit parts on the surface of the specimen do not change much, while the other parameters fluctuate. When the test time is long, the total volume and total surface area of the pits and protrusions on the surface of the specimen increase significantly, and the average depth of the pits and the average height of the protrusions both increase, which also indicates that with the increase of test time, the wear area on the surface of the specimen increases, the wear situation intensifies, and the surface damage accelerates.

**The influence of different particle sizes and concentrations on the coefficient of friction, temperature and wear**

In this test, in addition to observing the changes of the surface morphology of the specimens, the friction coefficient between the specimens, the changes in the temperature of the specimens and the wear data of the specimens before and after the test are measured by relevant equipment. Fig 19 is the change of friction coefficient between specimens

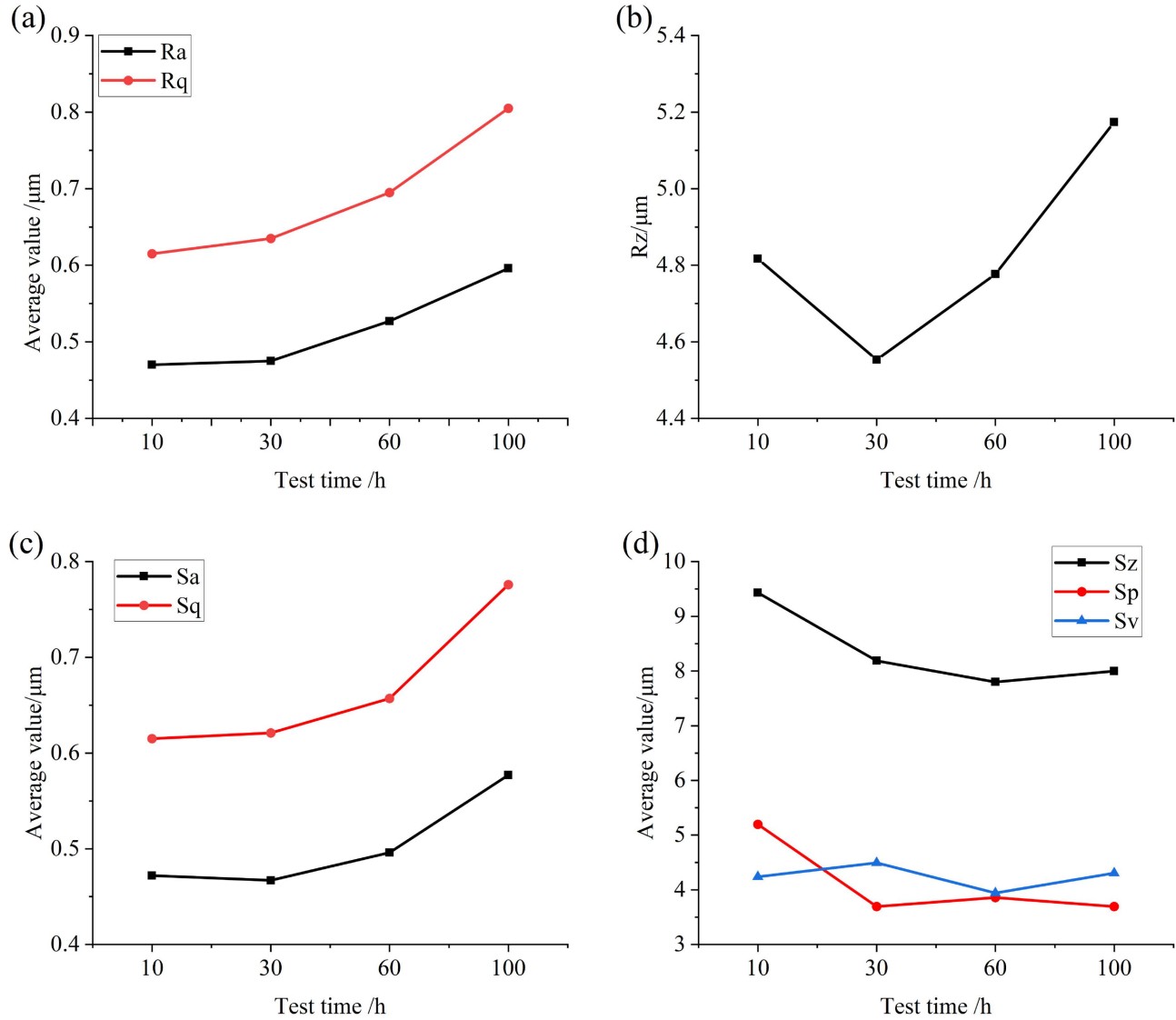

**Fig 16. Graphical representation of surface roughness values on the specimen surface after tests conducted at different time intervals (a) Ra, Rq (b) Rz (c) Sa, Sq (d) Sz, Sp, Sv.**

measured at different particle sizes at two test concentrations, and it can be seen that at the same concentration, with the increase of particle size, the friction coefficient generally shows an upward trend, and when the particle concentration increases, the corresponding friction coefficient also increases. This is due to the increase in the number of particles entering the friction interface and the increase in particle size, the degree of damage to the oil film formed between the friction surfaces increases, resulting in a gradual increase in the friction coefficient. Fig 20 is the temperature rise of different particle size test groups at two concentrations, it can be seen that with the increase of the particle size of the test particles, the temperature rise of the specimen increases correspondingly in the whole test process, and with the increase of particle concentration, the temperature rise data also increases slightly, but the improvement is not very obvious. This is because with the increase of particle size and concentration in the lubricating oil, the heat generated by friction between the specimens increases, which leads to a greater increase in the temperature of the specimen itself in the whole test

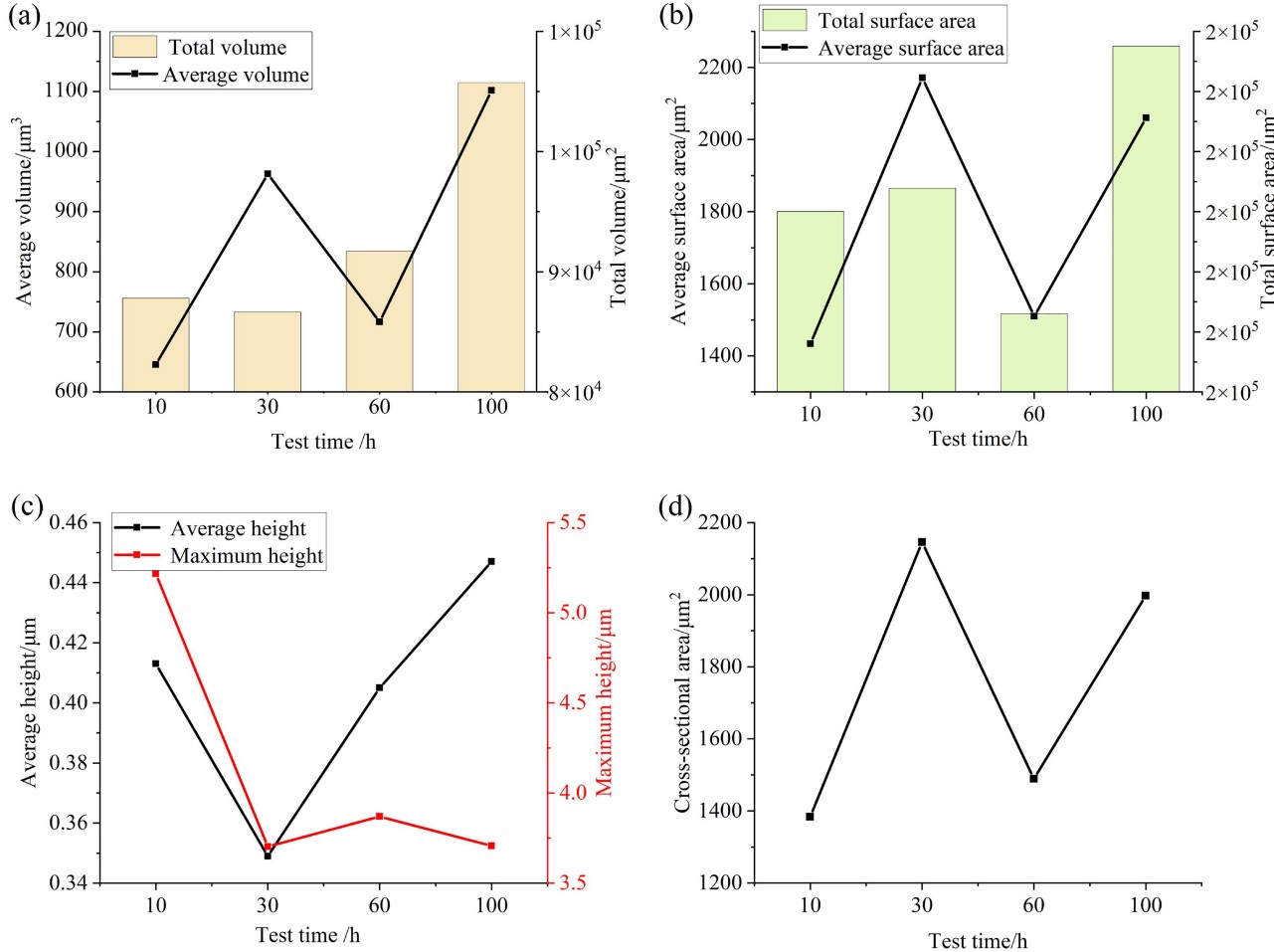

**Fig 17. Parameter diagram for raised features on the specimen surface (a) Volume (b) Surface Area (c) Height (d) Cross-sectional area.**

process, but because the temperature measurement adopts a non-contact temperature sensor, the accuracy of the measurement data is affected by the ambient temperature, so the temperature rise data in the figure fluctuates from time to time, but the overall trend is obvious, in line with the test law. Before and after the test, the quality of the toroidal specimen is measured with a precision balance, and the quality change diagram of the specimen obtained is shown in Fig 21, and it can be obtained from the figure that with the increase of particle size and concentration in the test, the wear amount of the specimen before and after the test is also gradually increasing, which is due to the more serious wear of the specimen caused by the particles of larger particle size and concentration, so that the amount of wear increases. However, in the test, there is also a negative amount of wear, which may be due to the compression and deformation of the particles under pressure between the specimens and thus adhere to the surface of the specimen, so that the quality of the specimen increases, or it may be caused by a certain measurement error of the balance itself.

## Conclusion

In this paper, the following conclusions are obtained by analyzing the surface damage mechanism of heavy-duty line contact parts induced by abrasive particles.

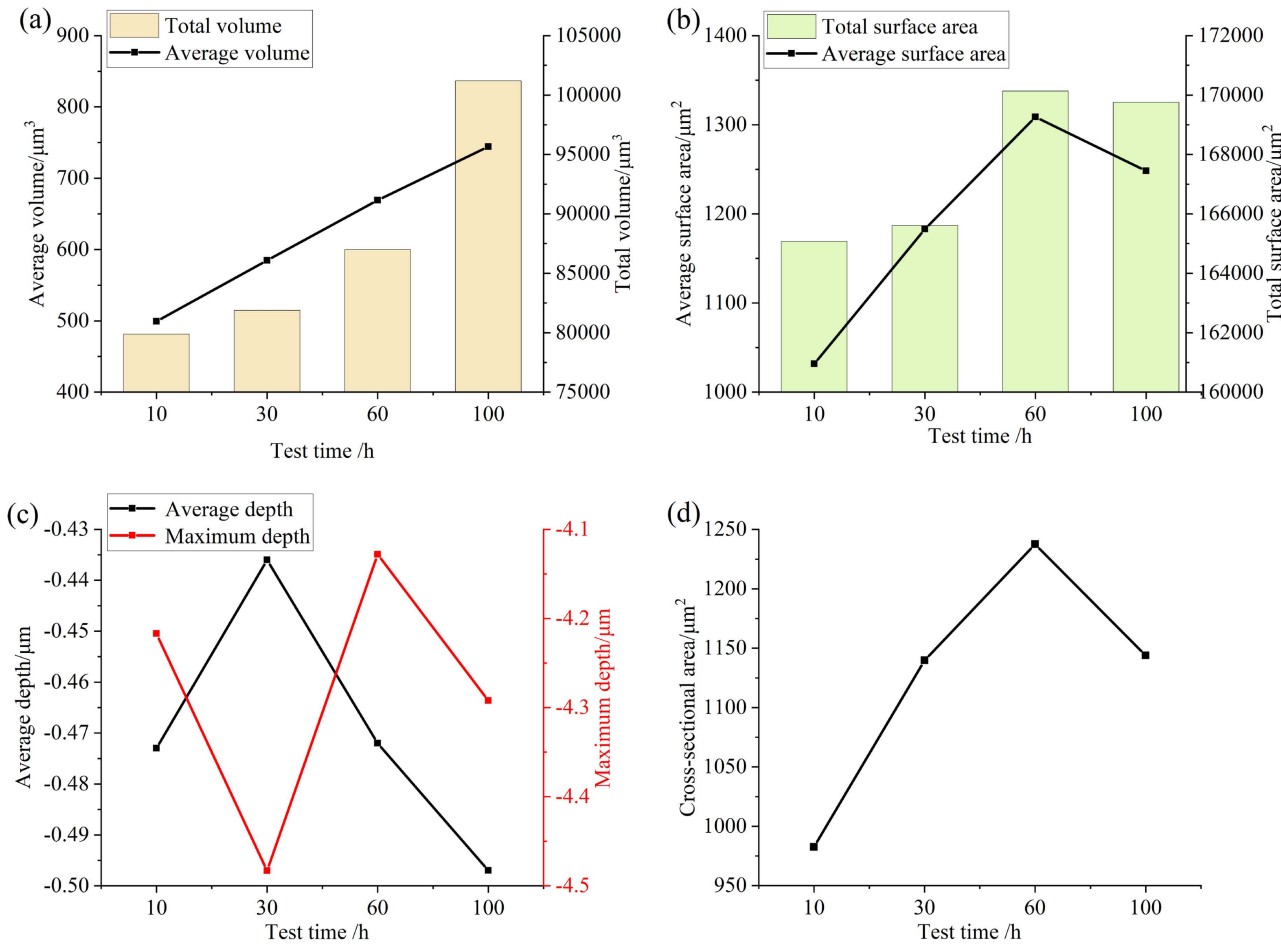

**Fig 18. Parameter diagram for pit features on the specimen surface (a) Volume (b) Surface Area (c) Depth (d) Cross-sectional area.**

(1)  With the increase of the particle size of iron oxide particles in the test, the two-dimensional roughness parameters and surface roughness parameters on the surface of the specimen decreased first and then increased after the test, and the related parameters of the convex part on the surface of the specimen also showed a tendency to decrease first and then increase, while the volume and number of surface pits increased with the increase of particle size. This is because when the particle size is small, the particles entering the friction surface will accelerate the wear of the micro-convex body on the surface of the specimen, and as the particle size continues to increase, the abrasive wear of the particles will be aggravated by the extrusion between the specimens, resulting in more and larger pits on the surface.

(2) When the concentration of iron oxide particles increases in the test, the wear on the surface of the specimen is aggravated, which is manifested by the increase of the two-dimensional and three-dimensional roughness parameters of the specimen surface and the increase of the volume, surface area and number of surface pits and protrusions. This is because when the concentration of iron oxide particles in the lubricating oil increases, more particles enter the friction surface, causing greater wear on the specimen surface.

(3) With the increase of the slip-roll ratio of the two specimens in the test, the roughness parameters of the surface of the specimen increase, and the total volume and total surface area of the surface protrusions and pits also increase.

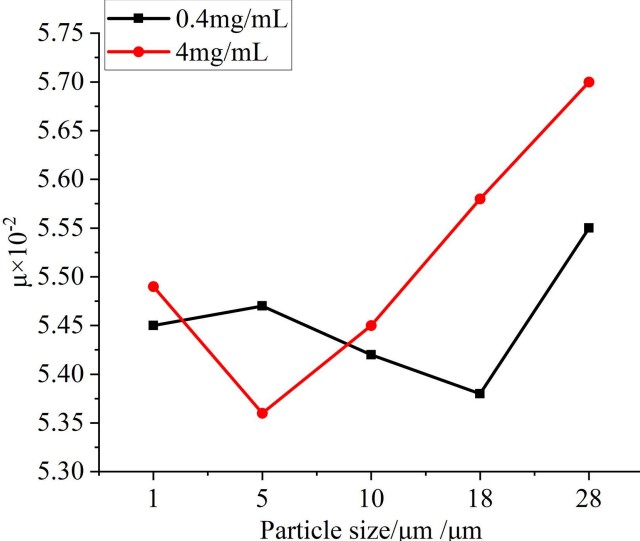

**Fig 19. Graph of friction coefficient variation for test groups with different particle sizes at two concentrations.**

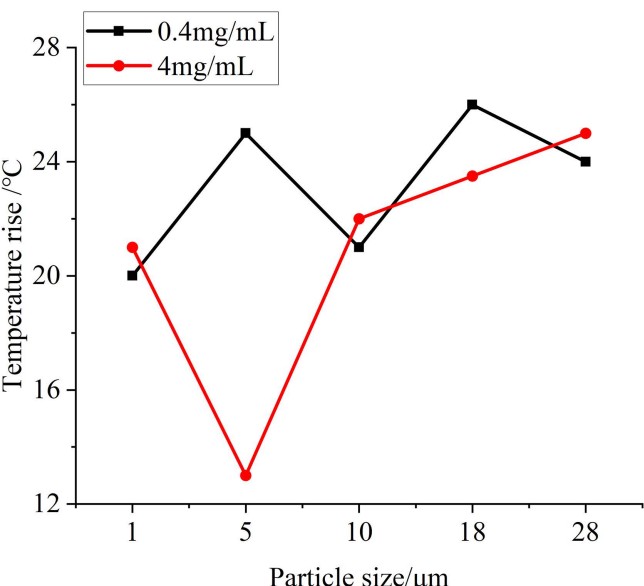

**Fig 20. Graph of temperature rise variation for test groups with different particle sizes at two concentrations.**

(4) When the test time increases, the wear on the surface of the specimen is more serious, and the wear particles doped into the lubricating oil increase, which leads to the increase of multi-line roughness parameters and surface roughness parameters on the surface of the specimen, resulting in a larger number of pits and protrusions, and the area and volume of these pits and protruding parts also increase with the increase of the test time.

(5) When the particle size and concentration of iron oxide particles in the lubricating oil increase, it will lead to the reduction of the friction performance of the lubricating oil, the increase of frictional heat generation, the aggravation of the

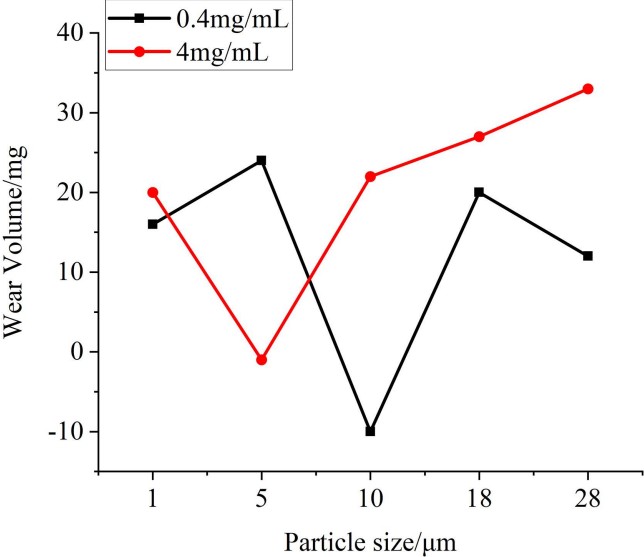

**Fig 21. Graph of wear volume variation for test groups with different particle sizes at two concentrations.**

surface wear of the specimen, and the increase of material loss, so that the friction coefficient between the specimens in the test increases, the temperature change of the specimen increases, and the wear of the specimen increases after the test.

## Supporting information

**S1 Data. Data Set.**
(XLSX)

## Author contributions

**Data curation:** Yang Li, Wei Wang.

**Formal analysis:** Wei Wang.

**Investigation:** Yang Li.

**Resources:** Wei Wang.

**Writing – original draft:** Yang Li.

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
