## [Editor Report · Decision Letter 0]

29 Aug 2025

Dear Dr. Li,

Thank you for submitting your manuscript to PLOS ONE. After careful consideration, we feel that it has merit but does not fully meet PLOS ONE’s publication criteria as it currently stands. Therefore, we invite you to submit a revised version of the manuscript that addresses the points raised during the review process.

We look forward to receiving your revised manuscript.

Kind regards,

Dr Pankaj Tomar

Academic Editor

PLOS ONE

Journal Requirements:

“National Natural Science Foundation of China (51875154)�Key Project of Natural Science Research in Colleges and Universities of Anhui Province (2024AH050503)�Anhui Provincial Key Research and Development Program (202004a05020057)”

Please state what role the funders took in the study. If the funders had no role, please state: 'The funders had no role in study design, data collection and analysis, decision to publish, or preparation of the manuscript.'

“This study is funded by National Natural Science Foundation of China (51875154)、 Key Project of Natural Science Research in Colleges and Universities of Anhui Province (2024AH050503)、Anhui Provincial Key Research and Development Program (202004a05020057)”

“National Natural Science Foundation of China (51875154)�Key Project of Natural Science Research in Colleges and Universities of Anhui Province (2024AH050503)�Anhui Provincial Key Research and Development Program (202004a05020057)”

5. We note that your Data Availability Statement is currently as follows: All relevant data are within the manuscript and its Supporting Information files.

**Additional Editor Comments:**

Kindly incorporate more references up to 50 for quantitative realization of paper.

---

## [Author Response · Author response to Decision Letter 1]

8 Sep 2025

In response to the editor's request, we have provided the following reply:

1. Requirement 1: Please ensure that your manuscript meets PLOS ONE's style requirements, including those for file naming.

Response We have revised this manuscript according to the formatting requirements of PLOS ONE. The modifications are marked in red and saved in the "Revised Manuscript with Track Changes" document.

2. Requirement 2 : Please note that PLOS ONE has specific guidelines on code sharing for submissions in which author-generated code underpins the findings in the manuscript. In these cases, we expect all author-generated code to be made available without restrictions upon publication of the work.

Response The experiments in this study were conducted on a JPM-1 friction and wear testing machine, with experimental data recorded in real time. The experimental data can be found in the attached file "Data set. excel".

3.Requirement 3: Thank you for stating the following financial disclosure: “National Natural Science Foundation of China (51875154)�Key Project of Natural Science Research in Colleges and Universities of Anhui Province (2024AH050503)�Anhui Provincial Key Research and Development Program (202004a05020057)”Please state what role the funders took in the study.

Response The role of the funder has been stated in the cover letter and highlighted in red. Please change the online submission form on our behalf. Thank you.

4. Requirement 4: About funding information

Response: Any funding-related text has been removed from the manuscript. The updated funding statement has been provided in the cover letter. Please change the online submission form on our behalf. Thank you.

5. Requirement 5: Please confirm at this time whether or not your submission contains all raw data required to replicate the results of your study. Authors must share the “minimal data set” for their submission.

Response: Our submission includes all the raw data necessary to replicate the findings of our study. The "minimal data set" has been uploaded as an attachment, named "Data set".

6. Requirement 6: If the reviewer comments include a recommendation to cite specific previously published works, please review and evaluate these publications to determine whether they are relevant and should be cited. There is no requirement to cite these works unless the editor has indicated otherwise.

Response: The reviewer did not make any suggestions in this regard.

7. Requirement 7: Please review your reference list to ensure that it is complete and correct.

Response: We ensure that the references are complete and accurate, with no citations to retracted papers.

---

## [Decision Letter · Decision Letter 1]

10 Oct 2025

Experimental study on the mechanism of surface damage induced by abrasive particles under heavy-load line-contact sliding-rolling conditions

PONE-D-25-45212R1

Dear Author

We’re pleased to inform you that your manuscript has been judged scientifically suitable for publication and will be formally accepted for publication once it meets all outstanding technical requirements.

Kind regards,

Pankaj Tomar

Academic Editor

PLOS ONE

Additional Editor Comments (optional):

Kindly include a few references (50 to 60) for academic rendering at large. Good luck

Reviewers' comments:

Reviewer's Responses to Questions

**Comments to the Author**

Reviewer #1: All comments have been addressed

Reviewer #2: (No Response)

Reviewer #3: (No Response)

2. Is the manuscript technically sound, and do the data support the conclusions?

Reviewer #1: Yes

Reviewer #2: Partly

Reviewer #3: Partly

3. Has the statistical analysis been performed appropriately and rigorously?

Reviewer #1: Yes

Reviewer #2: I Don't Know

Reviewer #3: Yes

4. Have the authors made all data underlying the findings in their manuscript fully available?

Reviewer #1: Yes

Reviewer #2: Yes

Reviewer #3: Yes

5. Is the manuscript presented in an intelligible fashion and written in standard English?

Reviewer #1: Yes

Reviewer #2: Yes

Reviewer #3: Yes

Reviewer #1: (No Response)

Reviewer #2: 1. The performance parameters of the lubricating oil should be specified. The viscosity of the lubricating oil directly affects the thickness of the oil film, and the film thickness is closely related to whether the abrasive particles could enter the frictional interface.

2. Fig.5: There are Chinese words in the picture. The data graphs only show the average values without labelling the error bars, (Fig.6, 7, 8, etc.).

3. Conclusion: As the size of the abrasive particles increases, the surface roughness decreases and then increases, and the volume and number of pits increases. There is a contradiction in this phenomenon. Roughness decreases and then increases because the surface tends to be flat and then becomes rough, and the increase in the number and volume of pits means that the surface damage is worsening. How can “surface flattening” and “continuous increase in pits” occur simultaneously on the surface morphology?

4. The negative wear in the experimental results was caused by the attachment of abrasive particles after deformation or by errors in balance measurements. Due to the specificity of this phenomenon, the authors should explain it further. To confirm the presence of abrasive particles, the component of abrasive particles on a surface could be detected by XRD or EDS.

5. Please refer to the journal's requirements to modify the format of diagrams and references.

Reviewer #3: 1. The novelty of the current work is unclear. The authors should clearly identify what new mechanism, relationship, or model is being revealed that advances understanding beyond existing literature.

2. Include a comparative discussion with previous works to show how this study extends or refines prior findings.

3. Please include the standard deviation to Figures 6-21.

4. The results section is highly descriptive but lacks quantitative interpretation or mechanistic reasoning. Please discuss the dominant wear mechanisms (abrasive wear, fatigue, micro-pitting) based on morphology and data trends.

5. The conclusions are mostly repetitive summaries. Please highlight the main findings.

---

## [Editor Report · Acceptance letter]

PONE-D-25-45212R1

PLOS ONE

Dear Dr. Li,

I'm pleased to inform you that your manuscript has been deemed suitable for publication in PLOS ONE. Congratulations! Your manuscript is now being handed over to our production team.

Kind regards,

on behalf of

Dr. Pankaj Tomar

Academic Editor

PLOS ONE